# Integrated GWAS and Transcriptome Analysis Reveal the Genetic and Molecular Basis of Low Nitrogen Tolerance in Maize Seedlings

**DOI:** 10.3390/plants14172689

**Published:** 2025-08-28

**Authors:** Fang Wang, Luhui Jia, Zhiming Zhong, Zelong Zhuang, Bingbing Jin, Xiangzhuo Ji, Mingxing Bai, Yunling Peng

**Affiliations:** 1College of Agronomy, Gansu Agricultural University, Lanzhou 730070, China; wangfang@gsau.edu.cn (F.W.); 13919055984@163.com (L.J.); zhuangzelong3314@gmail.com (Z.Z.); ff15293179783@163.com (B.J.); jixiangzhuo53@gmail.com (X.J.); baimingxing12@163.com (M.B.); 2State Key Laboratory of Aridland Crop Science, Lanzhou 730070, China; 3Jingning Agricultural Products Quality Safety Supervision and Inspection Station, Pingliang 743400, China; 4Institute of Geographic Sciences and Natural Resources Research, Chinese Academy of Sciences, Beijing 100101, China; zhongzm@igsnrr.ac.cn

**Keywords:** maize, low nitrogen, GWAS, RNA-seq, WGCNA

## Abstract

Nitrogen is an essential nutrient for the growth and development of maize (*Zea mays* L.), and soil nitrogen deficiency is an important factor limiting maize yield. Although excessive application of nitrogen fertilizer can increase yield, it can also cause environmental problems. Therefore, screening low-nitrogen-tolerant (LNT) germplasm resources and analyzing their genetic mechanisms are of great significance for the development of efficient and environmentally friendly agriculture. In this study, 201 maize inbred lines were used as materials. Two levels of low nitrogen (LN) (0.05 mmol/L, N1) and normal nitrogen (4 mmol/L, N2) were set up. Phenotypic indicators such as seedling length, root length and biomass were measured, and they were classified into LNT type (18 samples), nitrogen-sensitive (NS) type (27 samples) and intermediate type (156 samples). A total of 47 significant SNP loci were detected through a genome-wide association study (GWAS), and 36 candidate genes were predicted. Transcriptome sequencing (RNA-seq) analysis revealed that the differentially expressed genes (753 upregulated and 620 downregulated) in LNT materials under low nitrogen stress (LNS) were significantly fewer than those in NS materials (2436 upregulated and 2228 downregulated). Further analysis using WGCNA identified a total of eight co-expression modules. Among them, the red module was significantly correlated with root length and underground fresh weight under LN conditions (r = 0.75), and three key genes for stress response (*Zm00001d005264*, *Zm00001d053931*, *Zm00001d044292*) were screened out. Combined with GWAS, RNA-seq and qRT-PCR verification, eight candidate genes closely related to LNT at the seedling stage of maize were finally determined, involving biological processes such as stress response, nitrogen metabolism and substance formation. This study initially revealed the molecular mechanism of maize tolerance to LN through multi-omics analysis, providing a theoretical basis and genetic resources for breeding new nitrogen-efficient maize varieties.

## 1. Introduction

Nitrogen is a key nutrient element for the growth and development of maize, directly affecting its morphogenesis, photosynthesis and yield formation [1]. In agricultural production, although excessive application of nitrogen fertilizer has increased the yield of maize, it has also brought about problems such as environmental pollution and soil degradation [2]. Therefore, analyzing the genetic mechanism of efficient nitrogen utilization in maize and cultivating LNT varieties are of great significance for the development of resource-conserving and environment-friendly agriculture [3]. The absorption of nitrogen by plants mainly relies on two major systems: nitrate transporters (NRTs) and ammonium transporters (AMTs). In *Arabidopsis thaliana*, members of the NRT1 and NRT2 families (such as *AtNRT1.1* and *AtNRT2.1*) regulate the absorption and transport of nitrate nitrogen [4], while the AMT family (such as *AtAMT1.1-AtAMT1.5*) mediates the transmembrane transport of ammonium nitrogen [5]. The high expression of *OsNRT2.3* in rice (*Oryza sativa* L.) can significantly improve nitrogen use efficiency (NUE) and yield [6], indicating that nitrogen transport genes play a key role in crop nitrogen efficient utilization. In maize, the upregulation of gene expressions such as *ZmAMT1.3* and *ZmNRT2.1* is closely related to the transport of nitrogen to the aboveground part [7], *MADS26* affects nitrogen absorption and assimilation in maize by regulating the expression of genes related to nitrogen absorption and utilization [8], but the overall molecular mechanism still needs to be further explored. At present, there are various evaluation indicators for the LNT trait of maize, including root morphology (root length, root volume), biomass (above-ground/underground dry weight), photosynthetic parameters (SPAD value), and nitrogen metabolism enzyme activities, etc. [9,10,11,12]. However, due to the complex genetic background of nitrogen utilization, the screening criteria adopted in different studies have not yet been unified. In terms of genetic mapping, multiple studies have identified LNT-related loci in maize through QTL analysis. For instance, Li et al. located QTL clusters in the Bin 1.06–1.07 range. By enhancing the vertical tensile strength of the root system and the density of lateral roots, they significantly improved the nitrogen absorption efficiency in LN soil [13]. Guo et al. identified the main active QTL *qASI10* (Bin10.04), which shortened the silk spinning and powder release interval (ASI) without nitrogen application by 3.2 ± 0.4 days by regulating the expression of nitrate transporter protein (*ZmNRT2.3*), with a contribution rate as high as 50.28% [14]. Li et al. located *qNL1.1* (Bin1.06), which compressed the reduction in panicle length from 29.4% to 18.7% by maintaining nitrate reductase activity. The synergistic allele was derived from local germplasm Qi319 [15]. Zhang et al. found that *qSN3.2* (BIN3.04-3.05) regulates the auxin pathway gene *ZmARF16*, significantly reducing the coefficient of variation of the kernel row number in the lower spike under LN conditions and increasing the grain weight per spike [16]; He et al. identified 24 leaf type-related QTLS and 126 QTNs through a combined analysis of two populations at multiple nitrogen levels. They also discovered the LN-specific candidate gene *Zm00001d016468* (*NRT2*), which encodes a highly affinity nitrate transporter that significantly enhances the nitrogen absorption capacity of maize. Provide key targets for nitrogen-efficient and plant type collaborative breeding [17]. Li et al. based on transcriptome analysis of nitrogen-efficient/low-efficiency maize varieties, 34 hub genes such as *GRMZM2G076119* (bZIP transcription factor) under LNS were screened out, providing molecular markers for the breeding of LNT varieties [18]; Xing et al. integrated multi-omics data (genome/transcriptome/epigenome) to explore nitrogen-efficient gene resources in maize and multiple crops, identified key regulatory factors such as *ZmNLP7*, and developed SNP marker combinations that can be used for molecular design breeding [19]; Li et al. systematically analyzed the genetic and molecular mechanisms of maize nitrogen efficiency. Verified through CRISPR that *ZmNRG2.1* increased biomass by 21.3% under LN conditions by activating the expression of ammonium transporters, and proposed a “nitrogen signal-root configuration” synergistic regulation model [20]. In recent years, the combination of genome-wide association studies (GWAS) and transcriptome sequencing technology has provided new ideas for analyzing the genetic basis of complex traits. GWAS has been successfully applied to the gene mining of traits such as maize stem quality [21], drought tolerance [22] and root development [23], while transcriptome analysis (such as RNA-seq and WGCNA) can further screen key regulatory genes and co-expression networks [24,25,26]. However, most of the above-mentioned QTL studies have focused on agronomic traits at the mature plant stage, and due to the lack of cloning of candidate genes or the absence of dynamic mechanism analysis, functional genes have not yet been clearly identified. With the development of multi-omics technology, progress has been made in gene mining for maize tolerance to LN, but there are still limitations: (1) QTL mapping mostly focuses on agronomic traits at the mature plant stage, and the dynamic response mechanism at the seedling stage has not been clarified; (2) it is difficult to analyze the complex regulatory hierarchy in single-gene function studies. (3) multi-omics data have not yet been effectively correlated. In response to the above deficiencies, this study used 201 maize inbred lines as materials and combined phenotypic screening, GWAS, transcriptome analysis and WGCNA, aiming to (1) identify LNT and NS materials in the maize seedling stage, (2) explore key genes and pathways regulating LN tolerance, and (3) analyze the core genes in the co-expression network. The research results will provide molecular targets for nitrogen-efficient breeding of maize and lay the foundation for theoretical research on LN tolerance mechanisms.

## 2. Results

### 2.1. Phenotypic Variation Analysis and Material Screening

#### 2.1.1. Variation Characteristics of Phenotypic Traits Under Different Nitrogen Levels

The phenotype analysis of 201 maize inbred lines under two nitrogen levels (N1: 0.05 mmol/L; N2: 4 mmol/L) showed (Appendix A, Figure 1) that all measured traits exhibited continuous variation characteristics in the population. The LNT materials (294, 324) showed no significant changes in their properties at N1 and N2 nitrogen levels, while the nitrogen-sensitive materials (115, 285) showed more obvious differences. The average values of each trait under LNS (N1) were significantly lower than the normal nitrogen (N2) level. Among them, the seedling length, root length, aboveground fresh weight, underground fresh weight, aboveground dry weight and underground dry weight were 90%, 93%, 84%, 89%, 85% and 93% of the N2 level, respectively. From the perspective of variation characteristics, the coefficient of variation of aboveground dry weight was the largest (32.39%) at the N1 level, the coefficient of variation of underground dry weight was the largest (30.62%) at the N2 level, while the coefficient of variation of root growth was the smallest at the two nitrogen levels (N1:19.69%; N2:17.58%). The results of variance analysis showed that, except for root length and underground dry weight, the other four traits all showed extremely significant differences among genotypes (*p* < 0.001), and all traits were extremely significantly affected by nitrogen levels (*p* < 0.01). These results indicate that LNS significantly inhibits the growth of maize seedlings (especially with the most prominent effect on aboveground biomass accumulation), there are rich genetic variations in the population, and the interaction effect between genotype and nitrogen level is significant, suggesting that there are genetic differences in the response of different inbred lines to nitrogen stress.

#### 2.1.2. Correlation Analysis of Various Growth Indicators Among Associated Populations Under Different Nitrogen Levels

It can be seen from Figure 2 that at the LN level (N1): Most growth indicators show a significant positive correlation (for example, the r values of N1.RL with N1.LFW, N1.RFW, N1.SDW and other indicators all reach significant levels such as 0.353 and 0.644), and the correlation intensity is generally high (the r values are mostly in the range of 0.3–0.7). The synergy among growth indicators under LNS was further verified. At normal nitrogen levels (N2), there is a significant difference between the correlation pattern and LN. The correlation between some indicators significantly weakened (for example, the r value of N2.RL and N2.LFW was only 0.131, not reaching a significant level), and even a negative correlation occurred (for example, the r value of N2.RL and N2.SDW was −0.028); Meanwhile, some indicators still maintain a significant positive correlation (for instance, the r value of N2.RL and N2.SDW reaches 0.407 ***). This indicates that under normal nitrogen supply, the synergy pattern among growth indicators has been reconstructed, and the relationships among some indicators have shifted from “positive synergy” to “weak correlation” or “negative correlation”. At both N1 and N2 nitrogen levels, the correlations between LFW and SDW were extremely significant and of very high intensity (N1: r = 0.81 ***; N2: r = 0.82 ***). Combined with the scatter distribution of the corresponding subgraphs in the figure (the orange scatter points are highly linearly aggregated), the conclusion that “the biomass-related traits maintain a stable correlation pattern under different nitrogen environments” is further verified. This stability indicates that the association between aboveground fresh weight and aboveground dry weight has strong environmental adaptability in the changes of nitrogen levels and can be used as a reliable indicator of biomass traits in subsequent association analyses. In summary, the visualization results in the figure not only verify the previous conclusion that “there is a significant correlation between growth indicators at the two nitrogen levels and they meet the requirements of GWAS”, but also visually present the correlation pattern reconstruction of and the stability of biomass traits under the difference in nitrogen levels through a scatter plot. It provides a key basis for the selection of traits and the analysis of environmental effects in subsequent GWAS.

#### 2.1.3. Comprehensive Evaluation of LNT and Material Classification

A comprehensive evaluation is conducted by calculating the membership function values of LNT of each material. Subsequently, the membership function value was used to conduct a systematic cluster analysis on 201 associated groups, and the groups were classified into three categories (Appendix A). Among them, the first category was LNT materials (18 samples, accounting for 8.96%), which showed strong growth adaptability under low-nitrogen stress. The second category was intermediate materials (156 samples, accounting for 77.61%), which showed moderate sensitivity to nitrogen stress. The third category was NS materials (27 samples, accounting for 13.43%), which have significantly limited growth under LN conditions. This classification result provides an important basis for the subsequent screening of extreme materials and the research on the mechanism of LNT.

#### 2.1.4. Phenotypic Verification and Analysis of LNT and NS Materials

The phenotypic verification and analysis of 18 LNT materials and 18 NS materials indicated that the LNT materials showed stable performance of each trait at the two nitrogen levels (variation range < 15%), among which the variation of aboveground fresh weight was the greatest (CV > 35%), and the variation of seedling length was the smallest (CV < 25%). However, the growth of NS materials was significantly restricted under LN stress. The seedling length, root length, aboveground fresh weight and underground fresh weight decreased to 88%, 80%, 78% and 80% of the normal nitrogen level, respectively. The variation characteristics of their traits were manifested as follows: At the N1 level, the variation of aboveground fresh weight was the greatest (27.62%), and that of underground fresh weight was the smallest (20.97%). At the N2 level, the variation of underground fresh weight was the greatest (25.39%), and the seedling length was the smallest (16.2%) (Appendix A). This result confirmed the accuracy of the material screening. Both types of materials showed significant differences in nitrogen response characteristics and variation patterns of properties, providing reliable materials for subsequent research on the mechanism of LNT.

### 2.2. Genome-Wide Association Analysis

#### 2.2.1. Population Genetic Characteristics and Association Analysis Parameters

In this study, the Bonferroni correction method was used to determine that the significance threshold for genome-wide association analysis was *p* ≤ 1 × 10^−5^. Population genetic analysis indicates that this associated population has a relatively high genetic diversity (Figure 3A). Specifically manifested as 65% of individuals have a kinship coefficient between 0 and 0.1, and 27% between 0.1 and 0.2 (Figure 3B). The structural analysis based on the ADMIXTRUCT software (version 1.3.0) shows that the optimal subgroup number was k = 3, which can be clearly divided into three subgroups (P1, P2 and P3). This relatively simple group structure is conducive to reducing the false positive rate (Figure 3D). Linkage disequilibrium analysis revealed that at the r^2^ = 0.1 level, the average LD attenuation distance of the whole genome was 50–100 kb, and the LD attenuation characteristics of each chromosome were basically consistent with those at the whole genome level (Figure 3C). The above results indicate that the genetic structure of this population is reasonable and suitable for GWAS, providing a reliable genetic basis for subsequent research.

#### 2.2.2. GWAS Analysis Reveals the Distribution of Genetic Loci for LNT Traits in Maize

The mixed linear model (MLM) was used to analyze the LNT traits of maize. Based on the threshold *p* ≤ 1 × 10^−5^, a total of 47 significant SNP loci were detected at two nitrogen levels, and these loci were distributed across all 10 chromosomes (Figure 4). Among them, 25 significant loci were detected at the N1 level, 22 significant loci were detected at the N2 level, with the highest distribution on chromosomes 2 and 6 (a total of 15), 7 on chromosome 7, and the lowest distribution on chromosomes 1, 3 and 9. Specifically, a total of 15 significant loci for the seedling length trait were detected (12 at the N1 level, mainly located on chromosomes 2 and 7; 3 at the N2 level). Eight loci of root length were detected (3 at the N1 level; 5 at the N2 level); Seven loci were detected for underground fresh weight (6 at the N1 level; 1 at the N2 level). Seven loci were detected for underground fresh weight (4 at N1 level; 3 at N2 level). The *p* values of these significant SNP loci ranged from 3.17 × 10^−6^–9.56 × 10^−6^, showing strong statistical significance. It is worth noting that chromosome 7 shows a significant enrichment phenomenon (four loci) in the seedling growth trait, while chromosome 1 detects four significant loci in the underground fresh weight trait, suggesting that these chromosomal regions may contain key genes regulating the nitrogen response in maize. This result provides important genetic localization information for the subsequent mining and functional verification of candidate genes.

#### 2.2.3. Candidate Gene Prediction and Analysis

Based on GWAS (FDR < 0.05), a total of 47 significant SNP loci were identified. Through linkage disbalanced block analysis (D′ > 0.8, r^2^ ≥ 0.6), candidate regions were defined by extending 500 kb centered on significant SNPs, and stress response genes were screened in combination with the maize genome B73_RefGen_v4 annotation and MaizeGDB eQTL database. After functional filtering, 36 high-confidence candidate genes were finally obtained (Appendix A). Among them, 17 candidate genes were predicted under LN conditions (N1), and 20 candidate genes were predicted under normal nitrogen conditions (N2). The distribution of these genes on chromosomes has distinct characteristics: Chromosome 7 enriches the most candidate genes (five genes) at the N1 level, while chromosome 2 enriches the most (four genes) at the N2 level. Functional annotation analysis indicated that these candidate genes were mainly involved in the following biological processes: Cell wall/membrane biogenesis (46.12%), such as *Zm00001d021602* (a glycine-rich cell wall structural protein). Metabolic regulation, including *Zm00001d002589* (β-carotene hydroxylase), participates in photosynthesis, and *Zm00001d013048* (squalene synthase) regulates lipid metabolism. Stress response, such as *Zm00001d028347-49* (peroxidase family), participates in the oxidative stress response. Signal transduction, including *Zm00001d002675/78* (prolyl endopeptidase) and *Zm00001d006267* (ACT domain protein kinase). It is notable that multiple genes exhibit pleiotropic functions. For example, *Zm00001d017989* is simultaneously involved in amino acid transport, lipid metabolism and cell wall formation. *Zm00001d039650* (cytochrome P450) possesses both oxidoreductase and monooxygenase activities. Furthermore, some genes were detected under both nitrogen conditions (such as *Zm00001d053433*), suggesting their core role in the nitrogen response. These results provide important clues for analyzing the molecular mechanism of corn tolerance to LN.

### 2.3. Transcriptomic Analysis

#### 2.3.1. Transcriptome Sequencing Results and Quality Assessment

The results of the quality analysis of transcriptome sequencing data are shown in Appendix A. All samples obtained high-quality sequencing data. The number of raw reads ranged from 35,980,100 to 54,662,386. After strict quality control filtering, the number of clean reads reached 24,591,344 to 50,382,916. The quality assessment shows that the Q20 (sequencing error rate < 1%) and Q30 (sequencing error rate < 0.1%) indicators of all samples exceed 99.98% and 98.00%, respectively, indicating extremely high sequencing accuracy. Furthermore, the GC content was stable between 51.2% and 53.0%, consistent with the normal distribution range of the maize transcriptome.

#### 2.3.2. Analysis of Differentially Expressed Genes

Based on the screening criteria of log_2_FC ≥ 1 and *p* < 0.05, the leaves and root systems of different comparison groups were taken as the research objects, and and the total differentially expressed gene (DEG) analyses were conducted on LNT materials and NS materials, respectively (analysis and comparison: LN treatment N1 vs. normal nitrogen control). The results showed that in LNT materials, LNS (N1) induced significant upregulation of 753 genes and downregulation of 620 genes. A total of 2436 significantly up-regulated genes and 2228 significantly down-regulated genes were identified in NS materials. The total DEG number of NS materials is 3.4 times that of LNT, indicating that LNT materials may adapt to LN environments by maintaining gene expression homeostasis. Further analysis of shared DEGs revealed that the differentially expressed genes in response to the two types of materials accounted for only 18.7% of the total, suggesting that they might respond to NS through different molecular mechanisms. These results provide important clues for analyzing the molecular basis of LNT in maize.

Venn analysis was performed to identify common differentially expressed genes (DEGs) shared between leaves and roots of low-nitrogen-tolerant (LNT) and nitrogen-sensitive (NS) maize materials under two nitrogen levels (Figure 5). The results revealed 18 co-expressed genes between LNT leaves and NS leaves, 116 between LNT leaves and LNT roots, 42 between LNT leaves and NS roots, and 30 between LNT roots and NS roots. Notably, four core co-expressed genes, *Zm00001d033455*, *Zm00001d048950*, *Zm00001d048949* and *Zm00001d019060*, were shared across all four comparison groups, suggesting their fundamental regulatory role in maize’s response to low nitrogen stress (LNS). Functional annotation and homology analysis provide further insight into the potential mechanisms of these core genes: *Zm00001d033455* encodes a high-affinity nitrate transporter with high sequence similarity to *Arabidopsis thaliana AtNRT2.5* (NCBI accession: NP_195320), which is essential for nitrate uptake and remobilization under nitrogen deficiency. *Zm00001d048950* shows strong homology to *AtP5CS1* (NP_001107806) in *Arabidopsis*, encoding pyrroline-5-carboxylate synthetase (P5CS), a rate-limiting enzyme in proline biosynthesis that functions as both osmoprotectant and ROS scavenger during abiotic stress. *Zm00001d048949* is predicted to encode a glutaredoxin family protein with highest similarity to *AtGRXC7* (NP_001077303) in *Arabidopsis*, which participates in redox regulation and oxidative stress protection. *Zm00001d019060* is homologous to *AtbHLH93* (NP_001077303) in *Arabidopsis*, a transcription factor involved in iron homeostasis and chlorophyll biosynthesis pathways critical for nitrogen utilization and photosynthetic efficiency. These four genes may collectively contribute to LNS adaptation through coordinating nitrogen transport (*Zm00001d033455*), osmotic and oxidative balance (*Zm00001d048950* and *Zm00001d048949*) and transcriptional regulation of stress responses (*Zm00001d019060*). Their consistent co-expression across tissues and genotypes indicates a conserved regulatory network underlying nitrogen stress adaptation, while also reflecting tissue- and genotype-specific responses. This integrated mechanism offers potential targets for improving nitrogen use efficiency in maize breeding.

#### 2.3.3. Differential Gene GO (Gene Ontology) Analysis

GO analysis of differentially expressed genes in leaves: GO enrichment analysis based on normal nitrogen level leaf controls showed that LNT and NS materials exhibited different functional response patterns to LNS in leaf tissues. In LNT materials, differentially expressed genes are significantly enriched in three major functional categories: biological processes, cellular components and molecular functions. Among them, the biological processes are mainly involved in redox reactions, transmembrane transport and transcriptional regulation; the cellular components mainly involve membrane structure, membrane integration components and the nucleus; the molecular functions were most significant in terms of metal ion binding, transferase activity and hydrolase activity (Figure 6A). In contrast, NS materials have a lower degree of differential gene enrichment, and their biological processes mainly involve redox reactions, phosphorylation and carbohydrate metabolism; cellular components are concentrated in membrane structure, membrane integration components and cytoplasm; the prominent molecular functions are ATP binding, transferase activity and hydrolase activity (Figure 6B). These results indicate that LNT materials respond to LNS by activating a wider range of metabolic pathways and cellular processes, while NS materials exhibit relatively limited stress responses.

GO analysis of differentially expressed genes in root systems: GO enrichment analysis based on root systems with normal nitrogen levels showed that there were significant differences in the responses of LNT and NS materials to LNS in root tissues. The differentially expressed genes of LNT materials are mainly enriched in three major categories: biological processes, cellular components and molecular functions. Among them, the biological processes are mainly enriched in redox reactions, transmembrane transport and transcriptional regulation. The cellular components were significantly enriched in the membrane structure, membrane integration components and the nucleus. The molecular functions are prominently manifested as metal ion binding, transcription factor activity and hydrolase activity (Figure 6C). In contrast, NS materials exhibit a stronger transcriptome response, with a significantly greater number of differentially expressed genes than LNT materials. In biological processes, they mainly involve redox reactions, phosphorylation, protein phosphorylation and transcriptional regulation of DNA templates. The cellular components are also enriched in the membrane structure, membrane integration components and the nucleus. The molecular functions were most significantly characterized by transferase activity, ATP binding and metal ion binding (Figure 6D). These results suggest that although both types of materials activate basal metabolic pathways such as membrane-dependent and redox, NS materials exhibit stronger signal transduction and protein modification responses, suggesting that they may respond to LNS through more complex regulatory networks.

#### 2.3.4. KEGG (Kyoto Encyclopedia of Genes and Genomes) Annotation Analysis of Differentially Expressed Genes

The differential genes of each material were annotated through the KEGG database, and the metabolic pathways involved in the differential genes were classified, which could be divided into metabolism, membrane transport, signal transduction, cell cycle, etc.

Metabolic pathway analysis of differentially expressed genes in leaves: KEGG pathway enrichment analysis revealed the differential response mechanisms of leaves with LNT and NS materials to LNS (Figure 7). LNT materials are significantly enriched in photosynthesis—antenna proteins, plant hormone signal transduction and ribosome pathways (Figure 7A). NS materials are mainly enriched in nitrogen metabolism, porphyrin and chlorophyll metabolism, and amino sugar and nucleotide sugar metabolism pathways (Figure 7B). It is notable that the two types of materials jointly enriched 28 pathways, among which the pathways of plant hormone signal transduction, nitrogen metabolism and terpenoid backbone biosynthesis contained the largest number of differentially expressed genes. LNT materials are specifically enriched in 42 pathways. Besides the above-mentioned pathways, it also includes key metabolic pathways such as sphingolipid metabolism, photosynthetic carbon fixation and biosynthesis of unsaturated fatty acids. These results indicate that LNT materials adapt to the LN environment by maintaining the function of the photosynthetic system, regulating hormone signals and enhancing protein synthesis, while NS materials exhibit more fundamental nitrogen assimilation and chlorophyll metabolism responses.

Metabolic pathway analysis of differentially expressed genes in the root system: KEGG pathway enrichment analysis revealed (Figure 7) that there were significant genotype differences in the response of root tissues to LNS. LNT materials are mainly enriched in phenylpropanoid biosynthesis, cysteine and methionine metabolism, and glutathione metabolism pathways, such as metabolism (Figure 7C). However, NS materials exhibit a wider range of metabolic reprogramming, with higher enrichment in pathways such as phenylpropanoid biosynthesis, glutathione metabolism, starch and sucrose metabolism (Figure 7D). Two types of materials jointly enrich 66 pathways, among which phenylpropanoid biosynthesis, glutathione metabolism, starch and sucrose metabolism, plant hormone signal transduction, MAPK signaling pathway, cysteine and methionine metabolism, and nitrogen metabolism contain the largest number of differentially expressed genes. It is worth noting that LNT materials are specifically enriched in special metabolic pathways such as glucosinolate biosynthesis and biotin metabolism, which may be related to their unique LN adaptation mechanisms. These results indicate that although both types of materials activate basic stress pathways such as phenylpropane metabolism and antioxidant defense systems, LNT materials may enhance their tolerance to LNS by regulating the synthesis of specific secondary metabolites.

#### 2.3.5. Weighted Gene Co-Expression Network Analysis

Based on the transcriptome data of 24 RNA-seq samples, this study successfully constructed the weighted gene co-expression network (WGCNA). By conducting hierarchical clustering analysis on 10,672 highly expressed genes (FPKM ≥ 1), the network was constructed using the R software package WGCNA (version 1.72-1). The optimal soft threshold β = 12 is determined through the pickSoftThreshold function, and the constructed network conforms to the scale-free distribution characteristics. Using the dynamic shearing method for module division, eight feature modules were finally obtained through feature vector value calculation and module clustering analysis (height = 0.6), and the gene expression correlation patterns of each module are shown in Figure 8.

The module trait association analysis (Figure 8) showed that under LN conditions, the red module (r = 0.75) and green module (r = 0.6) were significantly positively correlated with underground fresh weight and root length, respectively; under normal nitrogen conditions, the yellow module (r = 0.65) is significantly correlated with aboveground fresh weight and seedling length. The identification of these key modules provides important clues for analyzing the molecular regulatory network of maize response to nitrogen stress, especially the red and green modules may contain key genes regulating root morphogenesis, while the yellow module may be closely related to aboveground growth and development.

The biological functional characteristics of the red module were revealed through GO and KEGG enrichment analysis (Figure 9). GO analysis showed that the module genes were significantly enriched in multiple important biological processes, including DNA template transcription regulation, protein phosphorylation and ubiquitination in signal transduction processes; membrane transport processes such as intracellular protein transport and vesicle-mediated transport; and stress response pathways such as redox processes. KEGG pathway analysis further indicates that these genes are mainly involved in key metabolic pathways such as ubiquitin-mediated protein hydrolysis, endoplasmic reticulum protein processing, peroxisome metabolism, transcriptional regulation and terpenoid skeleton biosynthesis. These results suggest that the red module may play an important role in maize root adaptation to LNS by regulating processes such as protein modification and degradation, oxidative stress response and secondary metabolism.

Based on the high correlation between the red module and LN traits, this study further screened the core regulatory genes of this module. By using Cytoscape 3.10.0 software to construct an interaction network for 97 genes within the module, it was found that Zm00001d005264 is located at the core of the network (Figure 10). Based on co-expression weight (weight > 0.2) and connectivity ranking, combined with the NCBI database and GO functional annotation, three key candidate genes were ultimately identified: Zm00001d005264, Zm00001d053931 and Zm00001d044292. Zm00001d005264 is a high-affinity nitrate transporter, with its top *Arabidopsis thaliana* homolog *AT1G08090* (NRT2.1) being crucial for nitrate uptake under low nitrogen conditions. Zm00001d053931, involved in protein folding and stabilization as a peptidyl-prolyl cis-trans isomerase, has *AT3G62030* (CYP20-3) as its closest homolog, which is known to participate in protein folding and stress response. Zm00001d044292, a glutathione peroxidase functioning in redox processes, is homologous to *AT4G11600* (GPX6) in Arabidopsis, which scavenges reactive oxygen species and maintains redox homeostasis under stress. These genes play important roles in pathways such as nitrogen absorption and transport, protein folding and stabilization and redox processes. They may participate in the formation of LNT traits in maize by regulating protein homeostasis and stress response. This result provides important target genes for understanding the molecular mechanisms of LNT in maize.

### 2.4. Candidate Gene Screening Based on Multi-Omics Integration

By integrating GWAS, upregulated differentially expressed genes specific to LNT materials (Appendix A), and WGCNA co-expression network analysis results, this study identified eight candidate genes closely related to LNT traits in maize (*Zm00001d053433*, *Zm00001d013046*, *Zm00001d024852*, *Zm00001d024843*, *Zm00001d012641*, *Zm00001d005264*, *Zm00001d053931*, *Zm00001d044292*). These genes have shown significant associations in various analyses, with *Zm00001d053433* and *Zm00001d013046* being significant in GWAS and differential expression analysis. *Zm00001d005264*, *Zm00001d053931* and *Zm00001d044292* exhibit high connectivity in the WGCNA network; *Zm00001d024852*, *Zm00001d024843* and *Zm00001d012641* are specifically upregulated in LNT materials. The functional annotations of these candidate genes indicate that they may be involved in nitrogen absorption and transport (*Zm00001d053931*), protein homeostasis regulation (*Zm00001d005264*), oxidative stress response (*Zm00001d044292*) and signal transduction pathways (*Zm00001d053433*). This multi-omics cross-validation strategy significantly improves the reliability of candidate gene screening, revealing gene functions from different levels and providing important targets for subsequent functional validation and molecular mechanism research.

### 2.5. Verification and Analysis of Candidate Gene Expression

qRT-PCR technology was used to validate the expression patterns of eight candidate genes (Figure 11). The results showed that the expression trends of all candidate genes (*Zm00001d053433*, *Zm00001d013046*, etc.) under LNS were highly consistent with RNA-seq data (*p* < 0.05), verifying the reliability of transcriptome sequencing results. Among them, *Zm00001d024852* and *Zm00001d012641* showed significantly stronger induced expression in LNT materials, with expression levels increased by 2.3 times and 1.8 times, respectively, compared to sensitive materials. The expression characteristics of these genes are consistent with their predictive functions, such as the NAC transcription factor encoded by *Zm00001d012641*, which may be involved in stress response regulation. This validation experiment not only confirmed the accuracy of multi-omics analysis results, but also revealed the expression characteristics of key genes through quantitative analysis, providing reliable molecular evidence and key research directions for subsequent gene function studies. Especially the significant differential expression of *Zm00001d024852* and *Zm00001d012641* suggests that they may be key candidate genes regulating LNT traits in maize.

## 3. Discussion

### 3.1. Identification Indicators and Screening System for LNT Materials

The genotype differences in crop nitrogen use efficiency (NUE) have become a research focus, and have been reported in crops such as rice [27], wheat (*Triticum aestivum* L.) [28] and rapeseed (*Brassica napus* L.) [29]. However, a unified NUE evaluation system has not yet been established, mainly due to the following reasons: (1) different evaluation indicators focus on different growth stages; (2) the traditional measurement of root morphology and physiological indicators is cumbersome. Through systematic analysis of previous studies, we found that morphological indices (e.g., plant height and biomass) proposed by Zheng [30] complement physiological parameters (e.g., nitrogen accumulation) employed by Cheng [31]. Notably, Li’s [32] study on the photosynthetic response mechanism of maize suggests that NUE evaluation requires comprehensive consideration of multidimensional indicators. Based on phenotype analysis of 201 maize inbred lines, we selected six sensitive indicators (seedling length, root length, aboveground/underground fresh weight and dry weight), among which aboveground fresh weight had the highest correlation with dry weight, which is consistent with the findings of Zeng [33] in rice. By using the membership function method, the materials were classified into three categories. The verification experiment showed that LNT materials exhibited a “compensatory growth” phenomenon in their root system under LN conditions (some materials increased root length by 5–15%), while NS materials showed significant reductions in various traits. This difference may be due to the following: (1) LNT materials having a stronger ability to distribute photosynthetic products to the root system; (2) NS materials experience more severe metabolic disorders under LN conditions. These results not only verify the reliability of the screening indicators, but also provide a theoretical basis for establishing a simplified evaluation system for LNT in the maize seedling stage.

### 3.2. Application of Genome-Wide Association Analysis in the Study of Maize Tolerance to Low Ni Trogen

Genome-wide association studies (GWAS), as a powerful tool for analyzing the genetic basis of complex quantitative traits, have shown unique advantages in crop genetic research [34]. This study is based on the principle of linkage disequilibrium (LD) [35] and conducted an association analysis of LNT traits using maize inbred line populations. A total of 53 significant SNP loci were identified, evenly distributed on 10 chromosomes. This result is highly consistent with previous studies on the distribution of nitrogen efficiency QTLs [36], confirming the multi-gene control characteristics of LNT traits in maize. It is worth noting that multiple significant loci discovered in this study exhibit co-localization with known agronomic trait QTLs: the seedling length related sites (chr1.S228644056, chr2.S19265414) overlaps with QTL chlorophyll content [37], indicating genetic coupling between plant morphogenesis and photosynthetic capacity; The aboveground fresh weight site (chr3.S227412930) was co-located with the QTL of kernels per row, and the seedling length (chr3.S192123462) was co-located with the QTL of kernel row number [38], confirming the synergistic regulation of biomass accumulation and yield formation. The root length-related loci (chr6.S155401982) and the panicle weight QTL are located in the same chromosomal region [39], which supports the breeding theory of “deep roots and large panicles” from a genetic perspective. The associated loci of underground fresh weight (chr6.S162546299) and the QTL that controls nitrogen efficiency are located in the same chromosomal region [40], providing anchor points for the analysis of the mechanism of nitrogen absorption and utilization. These co-localization sites form a trinity genetic network of “photosynthesis—source and sink—nitrogen efficiency”, laying the target foundation for molecular design breeding of LNT in maize (Appendix A). Notably, multiple genes predicted in this study have been confirmed to be involved in stress response: *Zm00001d053433* (E3 ubiquitin ligase) regulates stress response protein stability through the ubiquitin proteasome system [41]. *Zm00001d021019* (bHLH transcription factor) integrates multiple stress signaling pathways [42]. *Zm00001d039650* (cytochrome P450) regulates leaf greenness and substance metabolism [43]. These genes may constitute the core regulatory network of maize in response to LNS. Although this study did not find stable association loci across nitrogen levels, this is closely related to the genetic complexity of quantitative traits and environmental interaction effects [44]. In the future, by increasing marker density and expanding population size, it is expected to further elucidate the genetic mechanism of LNT in maize.

### 3.3. Analysis of the Transcriptional Regulatory Mechanism of Maize Tolerance to Low Nitrogen

The plasticity of maize root architecture under LNS is a key characteristic of nitrogen-efficient utilization [45]. This study found that LNT materials enhance nitrogen absorption capacity by promoting root growth (increasing total root length by 38% and underground fresh weight by 22%), while NS materials exhibit significant growth inhibition. This difference is consistent with the report by Zhang et al. [46], confirming the core role of root morphological plasticity in nitrogen uptake in maize. By comparing the transcriptome characteristics of LNT and NS materials, we found that the number of differentially expressed genes in roots was significantly higher than in leaves, indicating that roots are the main organ responding to LNS. Leaf differential genes are mainly enriched in photosynthesis-related pathways, while root genes are involved in metabolic processes such as phenylpropanoid biosynthesis. LNT materials respond to stress by activating basic metabolic pathways such as redox reactions and transmembrane transport, while NS materials exhibit more extensive transcriptional reprogramming. These results are consistent with Pan et al.’s [47] study on plant stress response, revealing the molecular basis of maize tolerance to LN.

Through weighted gene co-expression network analysis (WGCNA), this study identified the red module significantly correlated with root traits (r = 0.75). This module contains 12 core genes, among which the TPR protein encoded by *Zm00001d005264* (TPR protein) plays a key role in chlorophyll synthesis and stress response [48,49]. It may enhance the photosynthetic efficiency of plants under LN conditions by regulating the chlorophyll metabolism pathway, thereby improving plant stress tolerance. The peroxisome protein encoded by *Zm00001d053931* (peroxisome protein) regulates cell growth through the reactive oxygen species (ROS) signaling pathway [50,51]. Under LNS, ROS levels increase, and this gene may maintain cellular redox balance by clearing excess ROS, thereby protecting cells from oxidative damage. The ubiquitin protease encoded by *Zm00001d044292* (ubiquitin protease) is involved in nitrogen stress-induced protein degradation [52]. It may enhance the adaptability of maize to LNS by regulating the degradation process of proteins, optimizing the protein composition within cells, and improving nitrogen utilization efficiency. These genes form a collaborative regulatory network that determines the adaptability of maize to LNS by integrating stress signals, maintaining redox balance, and regulating nitrogen transport mechanisms. This discovery provides important target genes for further understanding the molecular mechanism of LNT in maize, and provides a theoretical basis for cultivating LNT maize varieties

### 3.4. Candidate Genes and Molecular Regulatory Networks for LNT in Maize

This study systematically elucidated the molecular mechanisms underlying maize responses to low nitrogen stress (LNS) through an integrated analysis of GWAS and WGCNA. Although the number of candidate genes identified via GWAS was limited—likely due to the stringency of the MLM model—their combination with WGCNA-based co-expression network analysis enabled the construction of a core regulatory network encompassing multiple key pathways (Figure 12). The reactive oxygen species (ROS) scavenging system emerged as a central component in maize adaptation to low nitrogen. Specifically, peroxidase *Zm00001d053931* (containing the PF00141 domain) and ascorbate peroxidase *Zm00001d044292* were found to function cooperatively in low nitrogen-tolerant (LNT) materials: the former mitigates oxidative damage through catalyzing H_2_O_2_ decomposition, while the latter sustains the reducing power within the ascorbate–glutathione cycle [53]. Ethylene and cytokinin were also identified as playing synergistic roles in regulating LNT. Under low nitrogen conditions, the ethylene synthase *Zm00001d024852* (PF14226 domain) promotes primary root elongation via a Fe^2+^-dependent reaction [54,55], while the cytokinin degradation enzyme *Zm00001d024843* (FAD domain PF01565) relieves suppression of lateral root development by reducing active cytokinin levels [56]. This “dual-track” model of root development—combining depth extension and lateral proliferation—enhances root surface area in LNT genotypes [57]. At the signal decoding level, the E3 ubiquitin ligase *Zm00001d053433* (RING domain PF13639) may modulate the stability of ZmNLP7 through ubiquitination, and the proteasome subunit *Zm00001d005264* (AAA-ATPase domain PF00004) helps maintain the degradation balance of nitrogen-responsive proteins [58]. Transcription factors also contribute critically to abiotic stress responses. For example, the bHL family member *Zm00001d030577* indirectly influences chlorophyll metabolism by regulating iron homeostasis. Although iron is not directly involved in chlorophyll synthesis, its role as a cofactor (e.g., for cytochrome P450 enzymes) is essential; iron deficiency disrupts chlorophyll synthesis in new leaves, chlorophyll transport in mature leaves, and overall photosystem functionality [59]. In this study, *Zm00001d030577* was significantly upregulated in LNT materials but suppressed in nitrogen-sensitive (NS) lines—consistent with the leaf yellowing and growth retardation observed in the latter. These findings not only reveal the molecular basis of maize low nitrogen tolerance but also construct a multi-tiered regulatory network encompassing redox homeostasis, secondary metabolism, hormone signaling and transcriptional control. This network offers a new perspective for investigating nitrogen use efficiency in maize and provides potential targets for molecular breeding. Future studies may validate the functions of these candidate genes through gene editing and further refine the molecular theory of LNT regulation in maize.

This study verified the expression patterns of candidate genes and constructed regulatory networks through qRT-PCR, but there are still three limitations: (1) Based only on transcriptional level data, there is a lack of direct evidence for protein translation and modification. (2) The functional mechanisms of genes need to be verified through genetic manipulation. (3) Field multi-environment tests have not yet been carried out. In the later stage, we will conduct CRISPR gene editing and phenotypic analysis, and carry out protein-protein interaction research and multi-environment field trials.

## 4. Materials and Methods

### 4.1. Experimental Materials and Phenotypic Analysis Under Low Nitrogen Stress

This study used 201 maize inbred lines as materials (provided by the Maize Research Group of Gansu Agricultural University, Appendix A). The seeds were disinfected with 10% H_2_O_2_ for 30 min, washed 3–4 times with distilled water, soaked for 24 h, and then transferred into seedling trays for cultivation in an artificial climate incubator(Top Cloud-Agri RTOP-310Y, Zhejiang, China) (day/night temperature 24/22 °C, relative humidity 60–80%, light exposure 14 h/day, light intensity 600 μmol·m^−2^·s^−1^). When the seedlings grow to the two-leaf stage, select 12 seedlings with uniform growth from each material and transfer them, respectively, into round plastic flowerpots for nitrogen treatment. Two nitrogen level treatments were set up in the experiment: LN treatment (N1, 0.05 mmol/L) and normal nitrogen treatment (N2, 4 mmol/L) [60]. Three biological replicates were set up in each treatment, and each replicate included four seedlings. A total of 30 mL of the corresponding nitrogen level nutrient solution was supplemented every 48 h. Morphological indicators were determined when the culture reached the four-leaf stage.

Determination of traits: Seedling height (length from the base of the stem to the top of the highest leaf), root length (length from the base to the end of the main root), biomass (fresh weight of the above-ground and underground parts was determined, respectively; After being blanched at 105 °C for 30 min, it was dried at 45 °C until a constant weight was reached, and the dry weight was measured).

The fuzzy membership function method [61] was adopted to evaluate the LNT of each material, and the specific calculation method is as follows:

(1) Calculate the relative value: *X_ij_* = LN treatment value/Normal nitrogen treatment value. (2) Standardization processing: *P_ij_* = (*X_ij_* − *X_jmin_*)/(*X_jmax_* − *X_jmin_*), where *X_jmin_* and *X_jmax_* are the minimum and maximum values of trait *j*, respectively. (3) Determine the weight coefficient: *W_j_* = *CV_j_*/∑*_j=_*_1_*CV_j_*, where *CV_j_* is the coefficient of variation of trait *j*. (4) Calculate the comprehensive LN tolerance index: *P_i_* = ∑*_j=1_*(*P_ij_* × *W_j_*), which reflects the comprehensive LNT of the *i*-th material. Its value ranges from 0 to 1, and the larger the value, the stronger the LNT.

### 4.2. Whole-Genome Resequencing and GWAS Analysis

The 55K SNP chip was used to perform whole-genome genotyping on 201 maize inbred lines (the sequencing data was provided by the Institute of Crop Sciences, Chinese Academy of Agricultural Sciences, independent biological repetition). After strict quality control (secondary allele frequency MAF > 0.05, genotype deletion rate < 20%), 558,529 high-quality SNP loci were finally obtained. The genetic structure of the population was analyzed using ADMIXTURE v1.3.0 software to determine the optimal number of subgroups. Meanwhile, based on SNP data, the kinship matrix (K matrix) was calculated using TASSEL v5.0 software, and the attenuation of linkage disequilibrium (LD) across the entire genome was analyzed. Genome-wide association analysis adopted the mixed linear model (MLM) in TASSEL v5.0, with population structure (Q matrix) and kinship (K matrix) as covariates, effectively controlling false positives. The significance threshold was set at −log_10_(P) ≥ 5. The Manhattan plot and QQ plot were plotted through the R package CMplot to visually display the distribution of significantly associated loci and the model fitting.

### 4.3. Transcriptome Sequencing and Data Analysis

The selection of transcriptome analysis materials based on the multi-index phenotypic evaluation system throughout the entire growth period was carried out. Firstly, a comprehensive evaluation was conducted by calculating the Membership Function Value of LNT of each material. Subsequently, the membership function value was used to conduct a systematic cluster analysis on 201 associated groups, and the groups were classified into three categories. Ultimately, four materials with the highest membership function values (GEMS59, Lx9801, Dan4245, GEMS2) were selected from the LNT category, and four materials with the lowest membership function values (Dong46, CIMBL16, CML304, CML115) were selected from the NS category as extreme phenotypic materials for transcriptome sequencing analysis. This clustering and extreme value screening strategy based on membership function values ensures that the selected materials have significant and stable phenotypic extremity in the population. The experiment set up two nitrogen levels (0.05 mmol/L and 4 mmol/L) for treatment, with a total of 8 treatment combinations:

NS materials: NS0.05L (leaves), NS0.05R (roots), NS4L (leaves), NS4R (roots)LNT materials: NT0.05L (leaves), NT0.05R (roots), NT4L (leaves), NT4R (roots)

Three biological replicates were set up for each treatment, and each replicate contained four seedlings. The material culture conditions are as follows: photoperiod of 14 h/10 h (light/dark), and diurnal temperature 24 °C/22 °C. Nitrogen treatment was carried out starting from the two-leaf and one-heart stage. A total of 30 mL of nutrient solution at the corresponding nitrogen level was irrigated every 48 h, and samples were taken at the four-leaf stage (about 20 days).

The total RNA of each sample was extracted by the TRIzol method. The library construction was completed by Lianchuan Bio Co., Ltd. (Hangzhou, China), and 150 bp double-ended sequencing was performed on the Illumina NovaSeq6000 platform. The original sequencing data were quality-controlled by Trimmomatic (version 0.39), and high-quality data (Q20 > 95%, Q30 > 90%) were retained. Clean reads were aligned to the maize B73 reference genome (v4 version) using HISAT2, and StringTie was used for transcript assembly and gene expression level (FPKM) calculation. Differentially expressed genes (DEGs) were screened using edgeR (version 3.40.0), with the threshold set as |log_2_FC| ≥ 1 and FDR < 0.05. The co-expression network was constructed using the R package WGCNA, and the genes with FPKM < 1 in each sample were filtered. The pickSoftThreshold function was used to determine the optimal soft threshold, and the co-expression module was constructed by the dynamic shearing method (minModuleSize = 30). Key hub genes were identified based on gene significance (GS) and module membership (MM).

### 4.4. Candidate Gene Verification and qRT-PCR Analysis

Based on the significant SNP loci (±50 kb interval) obtained from GWAS analysis, candidate genes were screened in combination with differentially expressed genes and WGCNA hub genes. cDNA templates were synthesized using the Evo M-MLV reverse transcription kit (Acrel Bio, Changsha, China) (Each RNA sample was independently reverse transcribed three times), and amplification detection was performed on the Illumina Eco real-time fluorescence quantitative PCR instrument using the SYBR Green Pro Taq HS premix kit (ROX Plus) (three technical replication Wells were set for each cDNA template). Taking the Actin gene as the internal reference, the relative expression level of the target gene was calculated by the 2^−ΔΔCt^ method. The primer sequences used are detailed in Appendix A.

### 4.5. Data Processing and Statistical Analysis

SPSS v25.0 software was used for analysis of variance (ANOVA) and Duncan’s multiple comparison test of phenotype data. The correlation analysis among traits was conducted using R v4.1.0 software. The gene co-expression network derived from WGCNA analysis was visualized using Cytoscape 3.10.0 to illustrate interactions among core genes.

## 5. Conclusions

This study established an LNT evaluation system for maize seedlings through multi-omics analysis, and screened out LNT and NS materials. LNT materials significantly enhance nitrogen capture efficiency by inducing compensatory root growth (such as increased root length and biomass), a process that is synergistically regulated by phenylpropane metabolism and REDOX balance pathways. This adaptive mechanism provides a theoretical target for breeding nitrogen-efficient maize varieties. Whole-genome association analysis identified 53 significant SNP loci, revealing the polygenic control characteristics of LNT traits. Transcriptome analysis and co-expression network analysis identified multiple core genes that jointly determine the adaptability of maize to LNS by regulating mechanisms such as redox balance, secondary metabolism and hormone signaling. This study has constructed a multilevel molecular regulatory network, providing a new perspective on the mechanism of nitrogen-efficient utilization in maize and potential targets for molecular breeding. In the future, candidate gene functions will be validated through gene editing to improve the molecular regulation theory of maize’s LNT.

## Figures and Tables

**Figure 1 plants-14-02689-f001:**
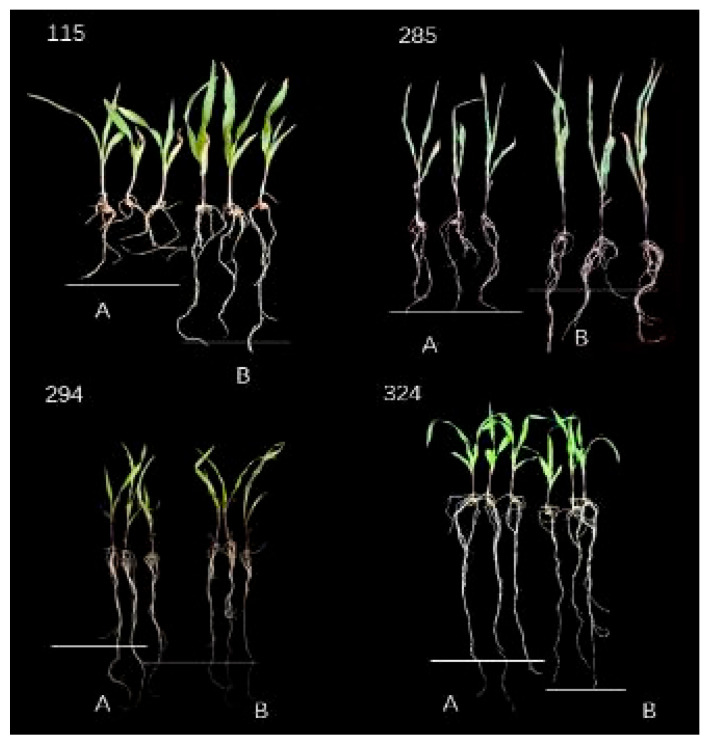
Phenotypic changes of some extreme materials under two nitrogen levels. Numbers 294 and 324 are LNT materials, while 115 and 285 are NS materials. A is LNS treatment (N1), the concentration is 0.05 mmol/L; B is normal nitrogen treatment (N2), the concentration is 4 mmol/L.

**Figure 2 plants-14-02689-f002:**
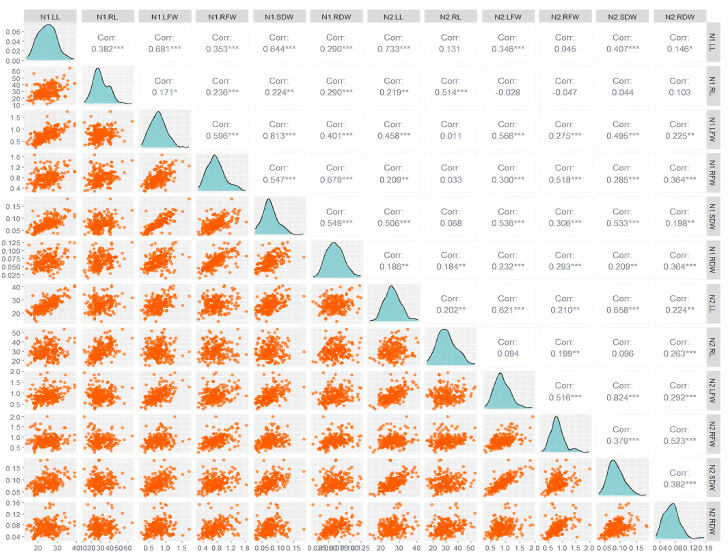
The phenotype distribution and correlation of the 6 traits of the associated group at the N1 (**left**) and N2 (**right**) level. LL: seedling length (cm); RL: root length; LFW: aboveground fresh weight; RFL: underground fresh weight; SDW: aboveground dry weight; RDW: underground dry weight; ns: no significant difference. *: 0.01 < *p* ≤ 0.05; **: 0.001 < *p* ≤ 0.01; ***: *p* ≤ 0.001.

**Figure 3 plants-14-02689-f003:**
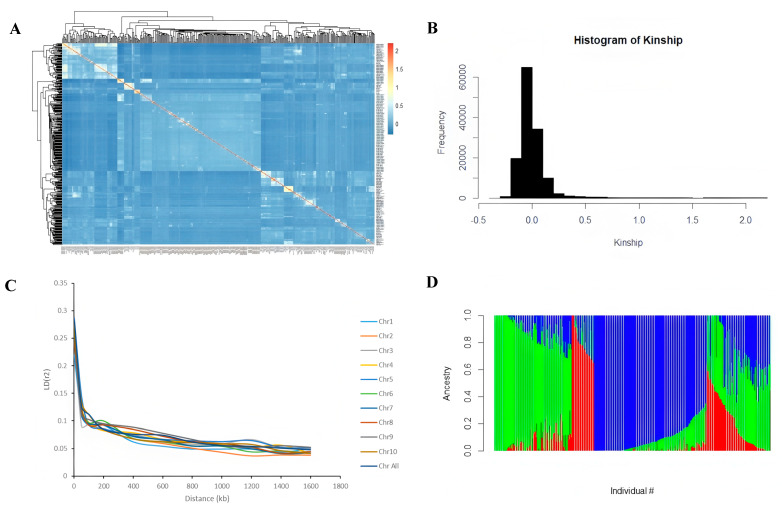
Genetic structure and linkage disequilibrium analysis of related populations. (**A**). Kinship heat map. (**B**). Linkage disequilibrium analysis. (**C**). Kinship distribution map. (**D**). Population structure. Each vertical bar represents an individual, and its color composition indicates the proportion of its ancestors coming from three different genetic backgrounds: P1 (blue), P2 (red), and P3 (green).

**Figure 4 plants-14-02689-f004:**
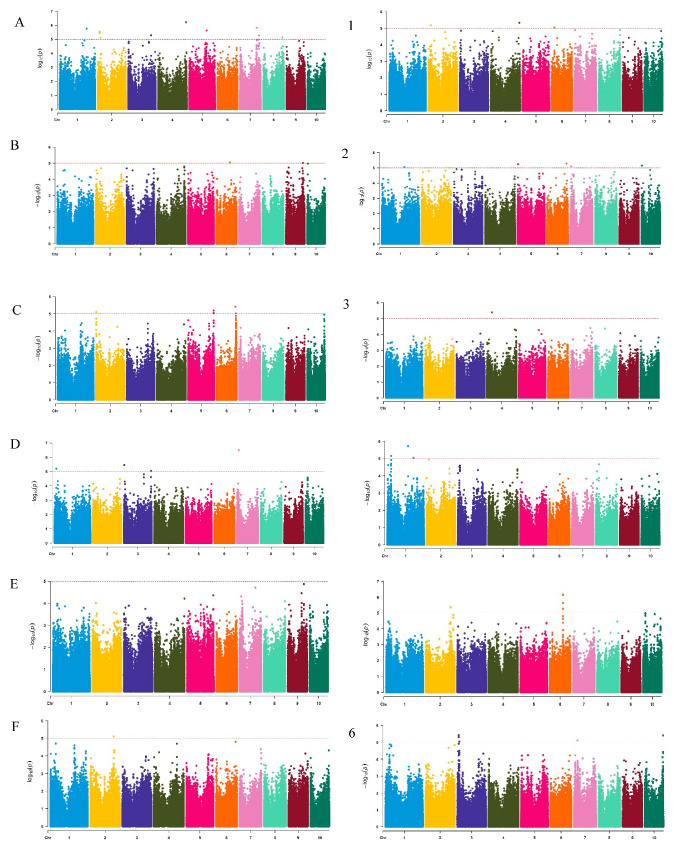
Manhattan plots of related traits in 201 corn-associated populations. (**A**), (**B**), (**C**), (**D**), (**E**) and (**F**), respectively, represent the seedling length, root length, aboveground fresh weight, underground fresh weight, aboveground dry weight and underground dry weight at N1 level; (**1**), (**2**), (**3**), (**4**), (**5**), (**6**), respectively, represent seedling length, root length, aboveground fresh weight, underground fresh weight, aboveground dry weight, and underground dry weight at N2 level. Data points are colored by chromosome (Chromosomes 1–10), with alternating colors distinguishing adjacent chromosomes. The red dashed line: genome-wide significance threshold (−log_10_(P) = 5, *p* ≤ 10^−5^).

**Figure 5 plants-14-02689-f005:**
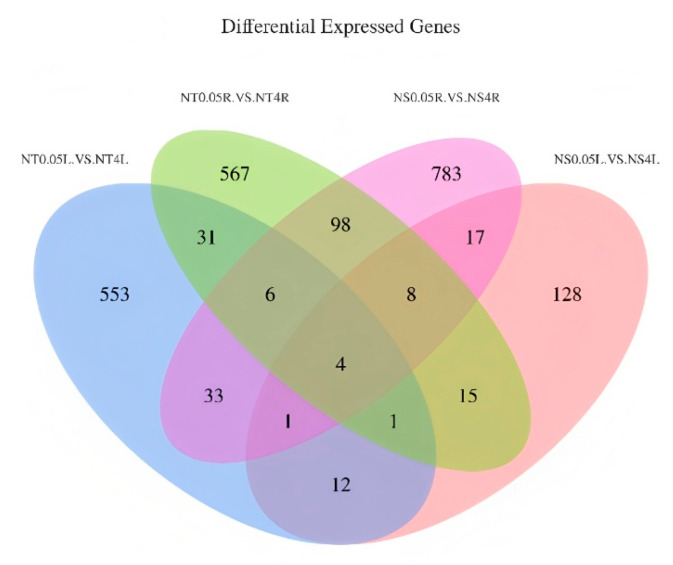
Venn diagrams of differentially expressed genes in different comparison groups. NT-0.05L vs. NT-4L: leaves of LNT material under 0.05 mmol/L treatment vs. leaves of LNT material under 4 mmol/L treatment; NS-0.05L vs. NS-4L: leaves of NS material under 0.05 mmol/L treatment vs. leaves of NS material under 4 mmol/L treatment; NT-0.05R vs. NT-4R: roots of LNT material under 0.05 mmol/L treatment vs. roots of LNT material under 4 mmol/L treatment; NS-0.05R vs. NS-4R: roots of NS material under 0.05 mmol/L treatment vs. roots of NS material under 4 mmol/L treatment.

**Figure 6 plants-14-02689-f006:**
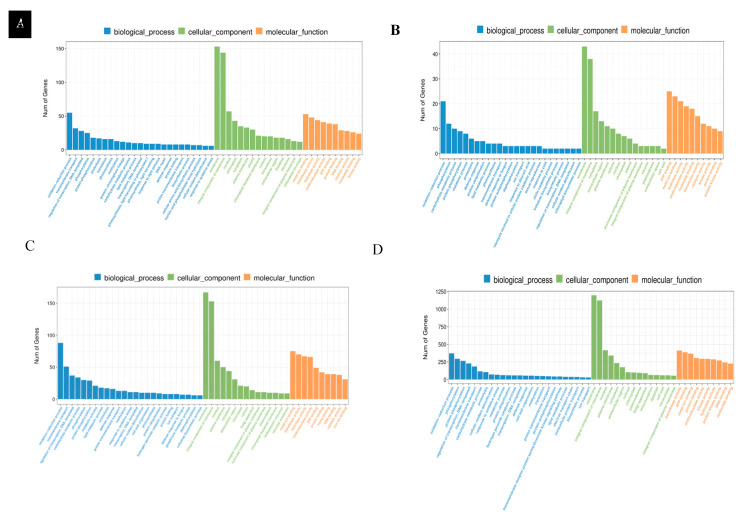
Differentially expressed gene GO function classification. (**A**). The GO functional classification of differentially expressed genes in the leaves of LNT materials under LNS. (**B**). The GO functional classification of differentially expressed genes in leaves of NS materials under LNS. (**C**). The GO functional classification of differentially expressed genes in roots of LNT materials under LNS. (**D**). The functional classification of GO differentially expressed genes in the roots of NS materials under LNS.

**Figure 7 plants-14-02689-f007:**
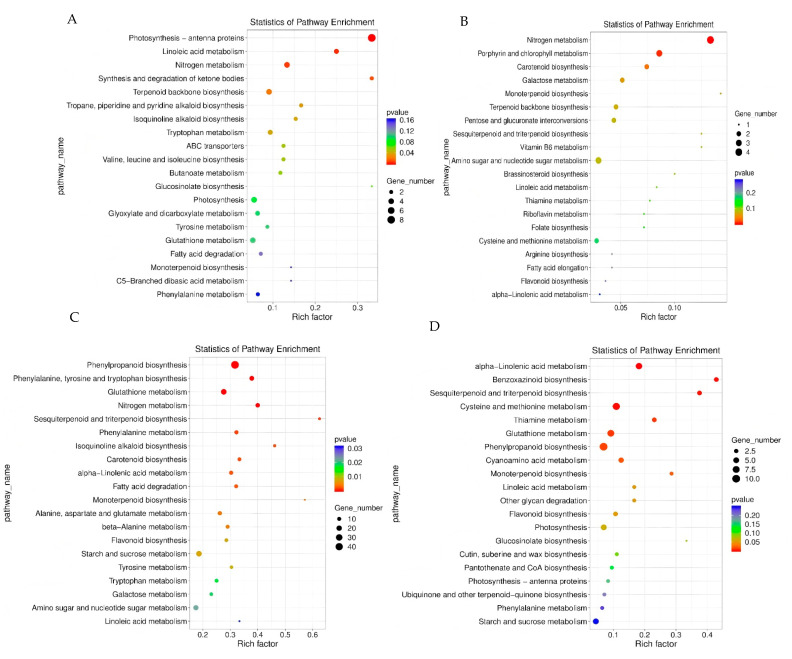
Functional classification of differentially expressed gene KEGG. (**A**) Leaves of LNT material under 0.05 mmol/L treatment vs. leaves of LNT material under 4 mmol/L treatment. (**B**) Leaves of LNT material under 0.05 mmol/L treatment vs. leaves of LNT material under 4 mmol/L treatment. (**C**) Roots of LNT material under 0.05 mmol/L treatment vs. roots of LNT material under 4 mmol/L treatment. (**D**) Roots of NS material under 0.05 mmol/L treatment vs. roots of NS material under 4 mmol/L treatment.

**Figure 8 plants-14-02689-f008:**
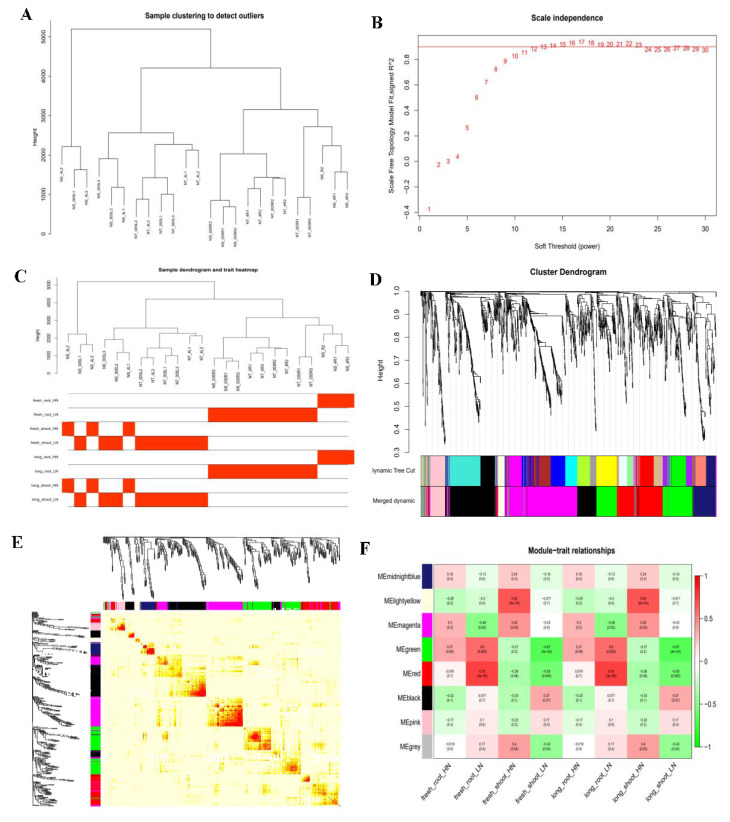
Sample clustering and gene module generation. (**A**). Sample clustering. (**B**). Scale independence. The red line represents the soft threshold and correlation coefficient corresponding to this WGCNA analysis. The higher the square of the correlation coefficient is, the closer the network is to a distribution without network scale. (**C**). Sample dendrogram and trait heatmap. The clustering tree is constructed based on sample similarity and highly reflects the distance between samples. The heat map is color-coded (red/white) to present the expression levels of different traits in each sample. (**D**). Gene network module. Different colors represent all modules, with gray indicatinggenes that cannot be classified into any module by default. (**E**). Gene co-expression network heat map. Different colors represent all modules, with gray indicatinggenes that cannot be classified into any module by default. (**F**). Heatmap of correlations between modules and traits. The leftmost color block represents the moduleand the rightmost color bar represents the correlation range. In the heatmap of the middle partthe darker the color, the higher the correlation, with red indicating positive correlation and green indicating negative correlation. The numbers in each cell represent correlation and significance. fresh_root: underground fresh weight; fresh_shoot: aboveground fresh weight; long_root: root length; long_shoot: seedling length; HN: normal nitrogen; LN: low nitrogen.

**Figure 9 plants-14-02689-f009:**
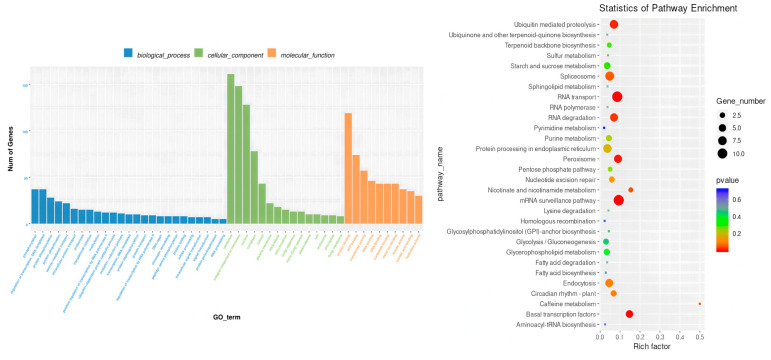
GO and KEGG annotations of genes in the red module.

**Figure 10 plants-14-02689-f010:**
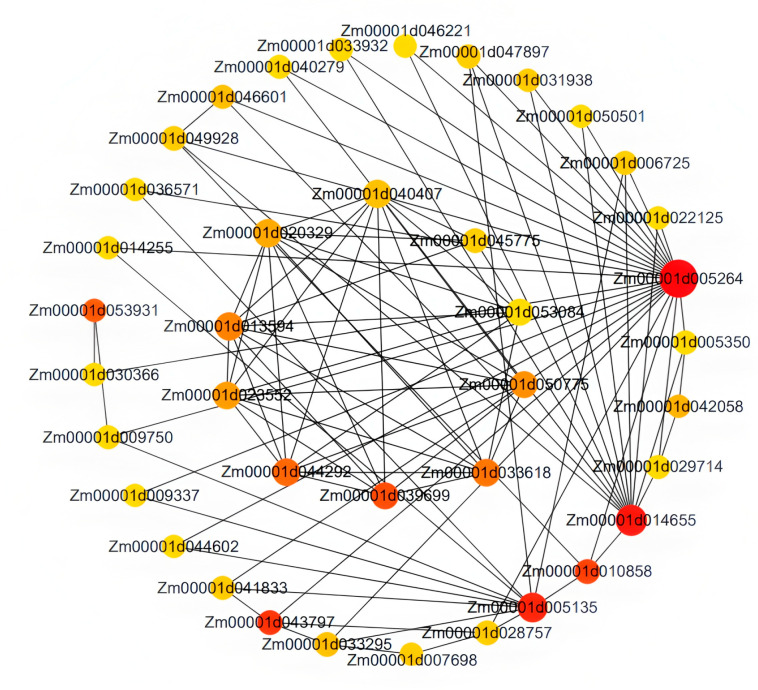
Gene co-expression network of the red module.

**Figure 11 plants-14-02689-f011:**
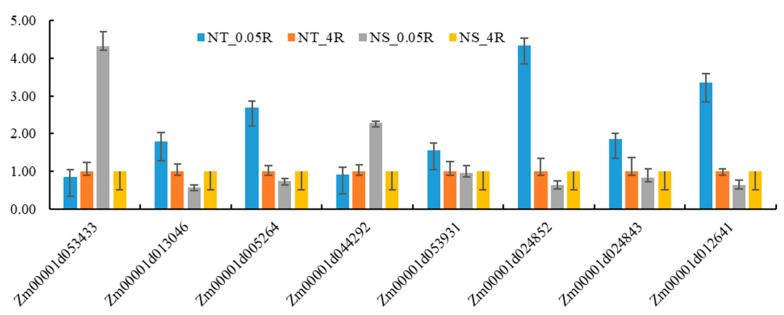
Candidate gene qRT-PCR verification. NT is a low nitrogen-tolerant material, NS is a nitrogen-sensitive material, 0.05R is a root system under low nitrogen stress, and 4R is a root system under normal nitrogen level.

**Figure 12 plants-14-02689-f012:**
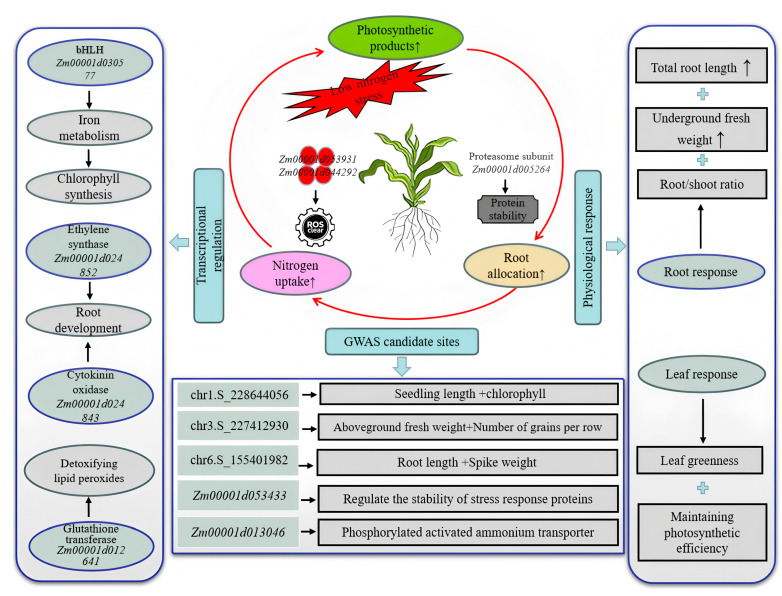
Multi-level regulatory network for low nitrogen tolerance in maize.

## Data Availability

Data is contained within the article and Appendix A.

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
