# Peer review of "Integrated GWAS and Transcriptome Analysis Reveal the Genetic and Molecular Basis of Low Nitrogen Tolerance in Maize Seedlings"

_plants, 2025, doi:10.3390/plants14172689_

Round 1
Reviewer 1 Report
Comments and Suggestions for Authors
The MS dentifying genetic and molecular bases for LNT using a combination of GWAS, RNA-seq, and WGCNA. The integration of multi-omics is a strength, and the identification of eight candidate genes provides novel insights. However, the paper requires further improvement in methodological clarity, statistical rigor, and data interpretation before it can be considered for publication.
1. Consider including a short comparative table summarizing prior LNT gene/QTL studies and highlighting the unique contribution of this work.
2. The criteria for classifying materials into LNT, intermediate, and NS types using the membership function method need more detail and justification. For example, why were six traits selected and not others (e.g., chlorophyll content, SPAD)?
3. More explanation is needed on how the "extreme phenotypes" were selected for transcriptome analysis (Section 5.3). Were these the top/bottom performers across all replicates or based on a composite index?
4. The MS reports 53 significant SNP loci but lacks Manhattan plot significance thresholds in the figure captions and main text.
5. Some SNPs are mentioned to co-localize with known QTLs. This is an important finding, but supporting references and a table mapping SNP positions to known QTLs should be included in MS.
6. The pipeline for integrating GWAS, RNA-seq, and WGCNA results into the final eight candidate genes should be clearly visualized (e.g., in a schematic flowchart). Currently, the integration steps are scattered across sections 2.2, 2.3, 2.5, and 2.4.
7. For each candidate gene, provide more functional evidence (literature references, domain analysis) to support its putative role in LNT.
8. The qRT-PCR validation of eight genes confirms expression patterns but does not enought.
9. Some figures (e.g., Manhattan plots, module-trait heatmaps) are difficult to read due to small font size. U had better to improve resolution.
10. Ensure that all supplementary tables/figures are properly cited in the main text in the order they appear.
11. State clearly the number of biological replicates used for each phenotypic measurement, GWAS dataset, and RNA-seq sample, and whether technical replicates were included.
12. For phenotypic data, provide error bars (±SE or ±SD) in figures to reflect variation.
13. Ensure consistent use of abbreviations (e.g., LNT, NS, LNS) throughout.
14. Update and expand references to include more recent studies (2022–2024) on nitrogen-use efficiency in maize and related crops.
Reviewer 2 Report
Comments and Suggestions for Authors
In considering the sustainability of food production, developing varieties that can produce high yields even under low-nitrogen conditions is becoming an important issue for various crops. This study evaluates low nitrogen stress tolerance in maize and is working to comprehensively elucidate its molecular mechanisms through GWAS, DEG analysis, and WGCNA analysis, which is expected to provide useful knowledge in a variety of fields. The research design and interpretation of the results were appropriate, and there are no major issues with publishing the paper. However, I would like to request the following improvements to make it easier for readers to understand.
1) p.3, Figure 1, it is unclear what characteristics these four types of materials exhibit. Further explanation is required in the figure legend or in the main text.
2) p.4, line 125, since Supplementary Table S2 is not included, the actual classification is unclear. What contributes to this classification into three groups? Also, is there any relationship between growth difference depending on nitrogen condition and biomass in the control? A more detailed analysis of the classification is required. By comparing the phenotypes in the low-nitrogen condition with the control, it may be possible to highlight the differences between the three groups.
3) p.4, Figure 2, correlations between traits are meaningful, but is it also necessary to check the correlations between different nitrogen conditions? In particular, the behavior of LNT and NS is thought to be an important clue in understanding the characteristics of them.
4) p.5, Figure 3, the figure legend is hidden by the figure and cannot be seen.
5) p.6, line 202, it is unclear how the authors narrowed down the candidate genes. The method for determining the LD block on which the detected SNPs are located should be explained.
6) p.7, in the RNAseq DEG analysis, the number of DEGs is compared between LNT and NS, but are these DEGs extracted only from the common DEGs in LNT and NS, respectively? or the summarized number of DEGs in LNT and NS? A clear description is required.
7) p.8, Figure 5, in line 236-237, a total of 1373 genes were altered in expression in LNT and a total of 4664 genes were altered in expression in NS, whereas in Figure 5, the total number of genes altered in LNT was 1329 and that in NS was 1106. Does this only show DEGs that are common to in LNT and NS, respectively? A clear description is required. If the authors had counted only the common DEGs, LNT would have shared 97% of the DEGs compared to only 24% in NS, highlighting an interesting difference between the two. It is possible that the genetic relatedness of the selected strains differs, but there may also be diversity in the mechanisms by which growth is retarded in LNS. This may be one of the reasons why GO pathway common to NS was limited in line 278.
8) p.8, line 264, is Figure 7 a mistake for Figure 6?
9) p.9, line 290, is Figure 8 a mistake for Figure 7?
10) p.11, line 384, is Figure 11 a mistake for Figure 10?
11) p.13, line 454, I cannot find Table 2 in the manuscript.
12) p.16, line 542-544, such information was not included in the results. The characteristics of the roots are also described in the Conclusion, so they should be presented systematically in the Results.
13) p.18, Figure 14, please cite appropriately in the main text.
14) This is not a question but a comment. In this study, the authors evaluated the low-nitrogen stress tolerance in relation to seedling growth, but I strongly hope that in the future they will evaluate whether this tolerance is actually useful for maintaining yield under low-nitrogen conditions.
Round 2
Reviewer 1 Report
Comments and Suggestions for Authors
Could be accept
Author Response
Thank you very much for taking the time to review this manuscript.